# Recent Advances in Sickle-Cell Disease Therapies: A Review of Voxelotor, Crizanlizumab, and L-glutamine

**DOI:** 10.3390/pharmacy10050123

**Published:** 2022-09-26

**Authors:** Michael Migotsky, Molly Beestrum, Sherif M. Badawy

**Affiliations:** 1Department of Medical Education, McGaw Medical Center, Northwestern University Feinberg School of Medicine, Chicago, IL 60611, USA; 2Department of Pediatrics, Ann & Robert H. Lurie Children’s Hospital of Chicago, Chicago, IL 60611, USA; 3Galter Health Sciences Library, Northwestern University Feinberg School of Medicine, Chicago, IL 60611, USA; 4Division of Hematology, Oncology, and Stem Cell Transplant, Ann & Robert H. Lurie Children’s Hospital of Chicago, Chicago, IL 60611, USA; 5Department of Pediatrics, Northwestern University Feinberg School of Medicine, Chicago, IL 60611, USA

**Keywords:** sickle cell, hemoglobinopathy, adults, adolescents, children, pediatric, crizanlizumab, voxelotor, l-glutamine, disease modifying therapy

## Abstract

Sickle-cell disease (SCD) is an inherited hemoglobinopathy, causing lifelong complications such as painful vaso-occlusive episodes, acute chest syndrome, stroke, chronic anemia, and end-organ damage, with negative effects on quality of life and life expectancy. Within the last five years, three new treatments have been approved: L-glutamine in 2017 and crizanlizumab and voxelotor in 2019. We conducted a literature search of these three medications, and of the 31 articles meeting inclusion criteria, 6 studied L-glutamine, 9 crizanlizumab, and 16 voxelotor. Treatment with L-glutamine was associated with decrease in pain crises, hospitalizations, and time to first and second crises, with a decrease in RBC transfusion rate. Barriers to filling and taking L-glutamine included insurance denial, high deductible, and intolerability, especially abdominal pain. Crizanlizumab was associated with a reduction in pain crises and time to first crisis, with reduction in need for opioid use. Adverse effects of crizanlizumab include headache, nausea, insurance difficulty, and infusion reactions. Voxelotor was associated with increased hemoglobin and decreased markers of hemolysis. Barriers for voxelotor use included insurance denial and side effects such as headache, rash, and diarrhea. These three medications represent exciting new therapies and are generally well-tolerated though price and insurance approval remain potential barriers to access. Other studies are ongoing, particularly in the pediatric population, and more real-world studies are needed. The objective of this article is to evaluate post-approval studies of crizanlizumab, voxelotor, and L-glutamine in SCD, with a focus on real-world efficacy, side effects, and prescribing data.

## 1. Introduction

Sickle-cell disease (SCD) is an inherited hemoglobinopathy, with an estimated 300,000 babies born worldwide with the disease [1]. In the United States, an estimated 100,000–120,000 people live with SCD, primarily of African American or Hispanic descent [2]. Due to a single amino acid substitution in the beta-globin chain, sickle cells are more prone to polymerization, vaso-occlusion, and hemolytic anemia in the setting of hemoglobin deoxygenation [3]. SCD is a chronic, lifelong disease characterized by acute and chronic problems, including painful vaso-occlusive episodes (VOEs), acute chest syndrome, stroke, chronic anemia, and end-organ damage in nearly all organ systems [4]. SCD also negatively affects patients’ health-related quality of life (HRQOL), with an inverse relationship to medication adherence [5,6]. While advances in treatment have led to an increase in life expectancy, models still show a reduced life expectancy of 54 years compared to 76 of controls [7].

First approved for SCD in 1998, hydroxyurea, which increases the fraction of hemoglobin F and thus decreases the likelihood of cells to deoxygenate and polymerize, remained the only U.S. Food and Drug Administration (FDA)-approved therapy for SCD for decades [8]. Hydroxyurea has shown to reduce vaso-occlusive episodes and other effects of SCD, leading to an increase in quality of life and morbidity and mortality [9]. However, adherence rates to hydroxyurea remain low, with estimates between 49% and 85% [10]. Over the past five years, three new medications have been approved for treatment of sickle-cell disease: L-glutamine (in 2017), crizanlizumab (in 2019), and voxelotor (in 2019).

L-glutamine, an amino acid, reduces oxidative stress of red blood cells, thus reducing endothelial adhesion [11]. In an initial Phase 3 double-blind placebo-controlled study, L-glutamine, both alone and in combination with hydroxyurea, was shown to reduce the number of pain crises compared to placebo [12]. Crizanlizumab, a P-selectin inhibitor, reduces adhesion between the endothelium and endothelial cells, platelets, sickled red blood cells, and leukocytes [13]. In a Phase 2 double-blind placebo-controlled study (SUSTAIN trial), crizanlizumab also resulted in a statistically significant reduction of pain crises [14]. Finally, voxelotor is a hemoglobin S polymerization inhibitor, increasing the cell’s affinity for oxygen and stabilizing the oxygenated hemoglobin state [15]. In its Phase 3 double-blind placebo-controlled study (HOPE trial), treatment with voxelotor significantly increased hemoglobin levels and reduced markers of hemolysis [16].

The objective of this article is to evaluate post-approval studies of crizanlizumab, voxelotor, and L-glutamine in SCD, with a focus on real-world efficacy, side effects, and prescribing data.

## 2. Literature Search and Articles Selection

A medical librarian (M.B.) performed a comprehensive literature search on 19 January 2022, of *PubMed*, *Embase*, *Cochrane Library*, *CINAHL*, *Scopus*, and *Web of Science*. The keywords “sickle cell anemia” AND “voxelotor” OR “crizanlizumab” OR “glutamine” were used, with a date range of 2017 to 2022, corresponding to the post-FDA-approval dates. This search returned 236 citations. Two citations were added through citation analysis. Following the initial collection, citations were evaluated based on the inclusion and exclusion criteria. Figure 1 depicts the flow of studies through the review.

### 2.1. Inclusion and Exclusion Criteria

Eligible studies included full articles and abstracts of clinical trials, including both the original and post-hoc/subgroup analyses of the Phase 2 and 3 studies cited above, retrospective studies, prospective studies, case series, and case reports. We included case series and case reports that captured any reported rare or uncommon side effects of any of these therapies. All were original studies of L-glutamine, crizanlizumab, or voxelotor on children or adults with SCD, with reports of clinical efficacy, side effects, or prescribing data. Review articles, molecular and pharmacokinetic studies, cost-effective models, studies prior to the initial Phase 2/3 studies, duplicates, or initial/interim data that were later reported more fully were excluded.

### 2.2. Data Extraction

The following data elements were extracted from the articles retrieved and meeting inclusion criteria: author name, year of study report, country of population studied, outcome measured, study design, age of study population, sample size, and a descriptive summary of findings related to clinical efficacy, side effects, and prescribing data.

## 3. Results

The literature search retrieved 236 citations, with two further articles added through bibliographies review. A total of thirty-one articles [12,14,16,17,18,19,20,21,22,23,24,25,26,27,28,29,30,31,32,33,34,35,36,37,38,39,40,41,42,43,44] met all inclusion criteria: six for L-glutamine [12,17,18,19,20,21], nine for crizanlizumab [14,22,23,24,25,26,27,28,29], and sixteen for voxelotor [16,30,31,32,33,34,35,36,37,38,39,40,41,42,43,44]. Twenty-four were conducted in the United States [12,14,17,18,19,20,21,22,23,24,25,26,27,28,29,35,36,37,38,39,40,42,43,44], while seven were conducted internationally [16,30,31,32,33,34,41].

### 3.1. Description of Included Studies

#### 3.1.1. L-glutamine

Of the six studies investigating L-glutamine, three stemmed from the same randomized control study, including the original multi-center Phase 3 study (*n* = 230) [12], a subgroup analysis evaluating outcomes based on the number of vaso-occlusive crises in the year prior to the study [18], as well as a post hoc analysis of umber of transfusions for the study group [17]. One case series (*n* = 4, ages 9–24) was included evaluating opioid use in patients starting L-glutamine [19]. Two single-center retrospective studies were included, both (*n* = 50 and *n* = 111) investigating prescribing data, compliance, and side effects of L-glutamine [20,21]. Table 1 summarizes the findings of these included studies.

#### 3.1.2. Crizanlizumab

Of the nine studies investigating crizanlizumab, four were from the original multi-center randomized-control SUSTAIN trial (*n* = 198), including the original Phase 3 data [14], and post hoc analysis of rates of and time to first hospitalizations [22], secondary endpoints [23], and days of opioid use [24]. One retrospective cohort study (*n* = 6) followed patients for 1 year following SUSTAIN, monitoring for pain crises, treatment patterns, and utilization of healthcare resources [25]. One pooled report between two Phase 2 trials (SUSTAIN and SOLACE-kids, *n* = 111) investigated safety and side effects of crizanlizumab [26]. One retrospective multi-center study (*n* = 297) evaluated insurance approval and adherence [28]. One retrospective review of a safety database (*n* = 28) evaluated infusion-related reactions [29]. One case report was included, describing two patients experiencing acute febrile reactions during crizanlizumab infusion [27]. Table 2 summarizes the findings of these included studies.

#### 3.1.3. Voxelotor

Of the sixteen studies investigating voxelotor, seven were related to the original randomized-control HOPE trial (*n*= 274), including the original Phase 3 data [16], post hoc analyses of hemolysis [30], leg ulcers [32], and concomitant use of hydroxyurea [41], and 72-week follow-up analyses of changes from baseline hemoglobin [31] and in VOE incidence [33] as well as long-term open-label extension (*n* = 178) monitoring hemoglobin response and markers of hemolysis [34]. One open-label, multicenter, multiple-dose trial (HOPE KIDS 1, *n* = 45) measured increase in hemoglobin and side effects in children [35]. Two Phase 2a studies evaluating increases in hemoglobin in adolescents were included at both 900 mg (*n* = 25) and 1500 mg (*n* = 15) per day [36,37]. Two single-center retrospective analyses were included investigating increases in hemoglobin (*n* = 17) and real-world experience of barriers to use and health outcomes (*n* = 54) [38,42]. Two multicenter retrospective chart reviews (*n* = 60 and *n* = 300) evaluated prescribing trends, side effects, and clinical efficacy [43,44]. One retrospective review (*n* = 1275) within one health system investigated the effect on anemia [40]. One pilot study (*n* = 9) evaluated effect on exercise capacity [39]. Table 3 summarizes the findings of these included studies.

### 3.2. Efficacy

#### 3.2.1. L-glutamine

In the original Phase 3 trial, treatment with L-glutamine vs. placebo led to a statistically significant reduction in pain crises, hospitalizations, time to first and second crises, ACS rates, and days in the hospital. The cumulative number of pain crises was 25% lower compared to the placebo group over the entire 48-week treatment period. It also showed no significant between-group differences in hemoglobin level, hematocrit level, or reticulocyte count or ED visits [12]. In subsequent analyses, pain crises rate ratios were similar in all subgroups when based on or adjusted for the number of crises in the year prior to the study period [18]. In a post hoc analyses of transfusion rates, patients had decreased units of transfusions per year in the treatment arm as well compared to placebo [17]. One case series showed patients treated with L-glutamine had decreased opioid usage without having changes in average hemoglobin [19].

#### 3.2.2. Crizanlizumab

In the SUSTAIN trial, treatment with higher-dose crizanlizumab showed a statistically significant reduction in the rate of vaso-occlusive episodes and longer time to first and second crises, with a decreased median rate of crises per year [14]. In post hoc studies of the SUSTAIN data, there was also a decrease in hospitalizations, increase in patient crisis-free time or days across multiple subgroups studied (i.e., higher baseline crises, HbSS genotype, concomitant use of HU), and reduction in days of opioid use [22,23,24]. In a 1-year follow-up from SUSTAIN, patients with higher-dose crizanlizumab had lower VOEs, but the study was underpowered to detect statistical significance. All patients still required opioids, one-third required transfusions, and all but one patient still required healthcare resources, such as emergency room visits or hospital admissions [25].

#### 3.2.3. Voxelotor

In the HOPE trial, treatment with voxelotor resulted in a statistically significant increase in hemoglobin level, with 51% of patients in the 1500 mg group having a response, with a mean change in hemoglobin of 1.1 g/dL (SD ± 0.13). This was also associated with a decrease in markers of hemolysis (i.e., indirect bilirubin and reticulocyte percentage). The annual incidence of VOE was not significantly different between the treatment and control groups [16]. In post hoc analyses of the HOPE data, those patients that had greater improvements in hemoglobin had the greatest reduction in markers of hemolysis, and nearly all patients receiving voxelotor in the trial had clinical improvement in their leg ulcers [30,32]. In 72-week follow-up of HOPE, the hemoglobin level increase was sustained, both with a mean increase in hemoglobin of 1.0 g/dL (SD ± 0.15) in up to 80% of patients, and it was associated with a decrease in the markers of hemolysis. They also had lower number of VOEs, but it was not statistically significant [31]. VOE incidence rates were lowest in patients who had achieved the highest hemoglobin levels, specifically higher than 10 g/dL [33]. In long-term extension of HOPE, the mean change in hemoglobin was 1.3 g/dL, with sustained decreases in markers of hemolysis [34]. Similar increases in hemoglobin of about 1 g/dL (ranging from 1–1.6) with decreases in markers of hemolysis were also found in the open-level multiple-dose trial HOPE KIDS 1, Phase 2a studies in adolescents, and multiple retrospective chart reviews [35,36,37,38,39,40,44]. Recent real-world data suggest potential clinical benefits related to reduction in VOE events, yet there are some inherit methodological limitations in these studies [40,43].

### 3.3. Quality of Life and Patient-Reported Outcomes Data

#### 3.3.1. L-glutamine

There have been no published data or studies related to the effects of l-glutamine on HRQOL outcomes in SCD.

#### 3.3.2. Crizanlizumab

There have been no published data or studies related to the effects of crizanlizumab on HRQOL outcomes in SCD.

#### 3.3.3. Voxelotor

Recent real-world data suggest that voxelotor was associated with improvement in HRQOL outcomes, as measured by the Clinical Global Impression of Change (CGI-C) and Patient Global Impression of Change (PGI-C) rating scales (Idowu, 2022 #6495). These observations are consistent with the CGI-C data reported earlier in the HOPE clinic trial.

### 3.4. Side Effects and Prescribing Data

#### 3.4.1. L-glutamine

In the original Phase 3 trial, adverse events were higher in the treatment group with l-glutamine compared to placebo, with nausea, chest, and musculoskeletal pain and fatigue being the most common side effects. About 2.7% of subjects discontinued therapy due to side effects [12]. In retrospective studies, since L-glutamine’s approval, most frequent side effects cited included abdominal pain, nausea, constipation, or taste of the medication [20,21]. When looking at l-glutamine prescribing data, at one pediatric center, 70% of prescriptions were dispensed, and 24% had good compliance (defined as ≥70%) [20]. At another adult center, at the end of a study period, 35% never filled the prescription, 42% discontinued, and only 19% were still actively taking L-glutamine. Barriers to initiating l-glutamine following prescription included denial of a prior authorization, high deductible, or other insurance issues. Reasons for stopping therapy included poor adherence, side effects, with a median day to stopping of 47 days [21].

#### 3.4.2. Crizanlizumab

In the original SUSTAIN trial, adverse effects occurring more often in the treatment crizanlizumab group compared to the placebo included arthralgias, diarrhea, pruritis, vomiting, and chest pain [14]. In pooled data among the SUSTAIN and SOLACE trials, 21.6% of patients suffered a serious adverse event, but only 5.4% were thought to be due to crizanlizumab. The most common adverse events reported included headache, nausea, and back pain [26]. Infusion-related reactions, such as pain or fever, were rare but cited, with a rate of 1.67 cases per 100 patient-years in a review of the safety database, and 1.8% having a reaction in the pooled trial data [26,27,29]. Infusion reactions usually occurred on the 1st or 2nd infusions, with most patients stopping treatment after developing a reaction [29]. Barriers to crizanlizumab included insurance denial and transportation issues to get the infusion [28]. Reasons for stopping, with 25–32% stopping treatment prematurely, included side effects, lack of feeling pain was improving, or that pain was worsening [26,28]. The use of premedication prior to crizanlizumab infusions varied, with 64% of patients not automatically using premedication [28].

#### 3.4.3. Voxelotor

The most common adverse events in the HOPE trial included headache, diarrhea, and nausea [16]. In subsequent studies, side effects were similar, with headache, diarrhea, nausea, and rash, usually mild, occurring in about a quarter of patients [35,36,37,38,42,44]. Discontinuation of the medication due to side effects was 0–23% of patients taking voxelotor [34,35,36,37,42,43]. About 15–29% of patients required dose adjustments due to side effects [42,43]. The most common reasons cited for prescription of voxelotor were reduction in anemia, VOEs, and pain [44]. However, in one review, 52% of patients prescribed medication did not start taking or did not follow-up [43]. In patients who were able to take voxelotor, the mean time from prescription to taking the medication was 45 days [42]. Barriers to prescription voxelotor included need for a prior authorization, lack of follow-up, or high co-pay [42]. A high proportion of patients were concomitantly taking HU, ranging from 52–100% [35,36,37,38,41,42,43].

## 4. Discussion

As the only new agents in decades, L-glutamine, crizanlizumab, and voxelotor represent exciting new therapies for SCD with potential clinical benefits to improve patients’ health outcomes. L-glutamine was found to have decreased pain crises, hospitalizations, opioid use, and rates of ACS but without affecting hemoglobin rates or markers of hemolysis. Side effects include nausea, pain, and fatigue but is generally well-tolerated. Crizanlizumab similarly resulted in decreased VOEs, hospitalizations, and opioid use, without major changes in hemoglobin. However, with it being an infusion, it has side effects in addition to headache, nausea, and back pain. Voxelotor was consistently found to increase hemoglobin levels and decrease markers of hemolysis. It is generally well-tolerated, with mild side effects related to headache, GI symptoms, and rash.

All three medications have significant cost associated with them, with insurance approval and co-pay of the medication and, for crizanlizumab, transportation to the infusions as barriers to care. However, in a cost-effective analysis for L-glutamine, treatment alongside HU resulted in a decrease in overall cost due to hospitalizations averted [45]. Similarly, for crizanlizumab, whereas crizanlizumab alone was not cost-effective, when added to HU, the decrease in VOEs and hospitalizations made crizanlizumab more cost-effective than the cost of the medication itself [46]. There is growing evidence for real-world data of all these medications, which would be key to optimize their utilization and be able to overcome barriers and other logistical considerations. Moreover, taking a patient-centered approach is critical to better understand how to select the right disease-modifying therapy that would benefit patients the most based on their clinical phenotype and genotype.

Ongoing trials are further looking into the effects of these medications beyond the initial pivotal studies. The STEADFAST trial is investigating crizanlizumab versus standard of care, with respect to chronic kidney disease (CKD) and SCD nephropathy [47]. In addition, while some of the trials included adolescents, further studies are now looking further into the pediatric population, including crizanlizumab in the STAND and SOLACE-Kids trials [48,49]. The HOPE KIDS 1 and 2 studies are investigating voxelotor’s effects on pediatric patients as well as its specific effect on TCDs [50,51]. It is worth noting that voxelotor was recently FDA-approved for children aged 4–11 years old based on clinical benefits related to hemoglobin level increase. There also are multiple other therapies in the pipeline with promising early data, such as inclacumab (p-selectin inhibitor), GBT021601 (next-generation HbS polymerization inhibitor), etavopivat and mitapivat (pyruvate kinas activators), FTX-6058 (fetal hemoglobin induction), IMR-687 (inhibitor of phosphodiesterase 9), and CSL889 (plasma-derived hemopexin), among others. There are several other active studies related to curative approaches for SCD, such as gene therapy and gene editing, which have shown encouraging results so far; yet, long-term follow-up is still needed to ensure sustained clinical benefits and favorable safety.

Strengths of this literature review include its broad and thorough approach, including not just established and fully published clinical trials but also abstracts of retrospective studies and case reports, many of which published within the last year, which allows it to be more up to date on the most recent data coming out on these therapies. It also included not just clinical effects but also prescribing data and real-world data, which are helpful and practical for patient care.

Limitations of this review include the fact that several of the included studies stem from the original three pivotal trials for the medications, with many publications having post hoc analyses or limited single-center chart review studies. Additionally, many of the most recent studies are still abstracts, without full trials finishing up since the approval of the medications. Finally, there have yet to be many publications on the effects of these medications on other effects of SCD beyond VOE, such as end organ damage and health-related quality of life.

In conclusion, SCD a lifelong disease with risk for several complications and significant impairment of quality of life. These three disease-modifying therapies, namely l-glutamine, crizanlizumab, and voxelotor, represent the start of a new era for novel therapies for SCD with several others in the pipeline SCD. L-glutamine and crizanlizumab have both shown impact on reducing VOEs and days in the hospital, with voxelotor improving anemia-reducing hemolysis. All three therapies are generally safe and effective for SCD patients with potential clinical benefits, and they are considered valuable additions to our treatment arsenal, especially when hydroxyurea is not well-tolerated or not desired by patients or caregivers.

## Figures and Tables

**Figure 1 pharmacy-10-00123-f001:**
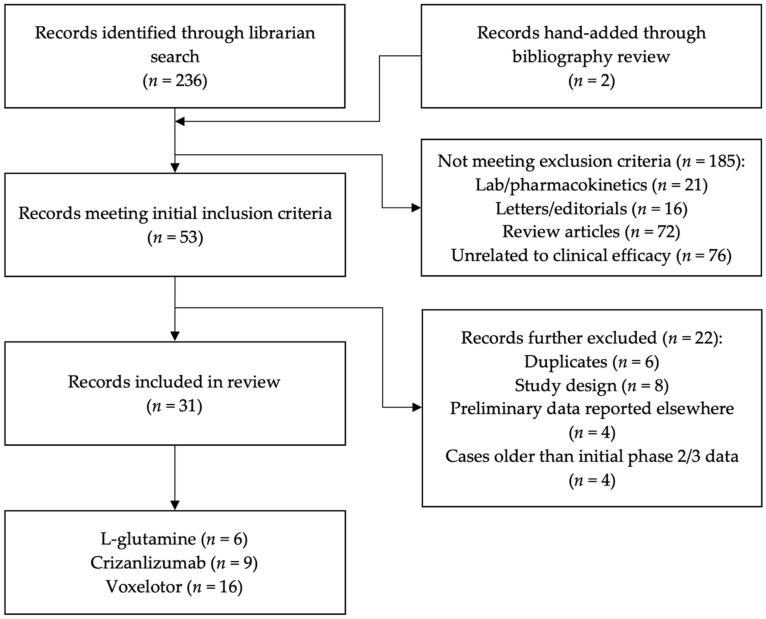
Flow of studies through selection process.

**Table 1 pharmacy-10-00123-t001:** Summary of findings from included studies of L-glutamine (Endari).

Source (Country)	Outcome Measured	Study Design	Age, Years	Group, *n*	Summary of Findings
*Clinical Efficacy*					
Niihara 2018, USA	Primary: number of pain crises; secondary: hospitalizations, ED visits, changes in hematologic measures	RCT	5–58, median 19 in treatment group, 17 in placebo	230	- Pain crises: median 3 in treatment group, 4 in placebo (*p* = 0.005) - Hospitalizations: median 2 in treatment, 3 in placebo (*p* = 0.005) - ED visits: did not differ significantly (1 in both) - No significant between-group differences in the change in hemoglobin level, hematocrit level, or reticulocyte count - Cumulative number of pain crises was 25% lower in the l-glutamine group than in the placebo group over the entire 48-week treatment period - Time to first pain crisis: 84 vs. 54 days (HR 0.69, *p* = 0.02) - Time to second crisis: 212 vs. 133 (HR 0.68, *p* = 0.03) - ACS: 8.6% vs. 23.1%, *p* = 0.003 - Days in hospital: 6.5 vs. 11, *p* = 0.02 - Adverse events higher in placebo vs. L-glutamine (100% vs. 98.0%), as was serious adverse events (87.1% vs. 78.2%)
Niihara 2018, USA	Number of crises	Subgroup analysis of Niihara et al.	5–58, median 19 in treatment group, 17 in placebo	230	- Rate ratios of pain crises similar in all subgroups (based on # of crises in year prior to study): 0.87, 0.74, and 0.82 for 2, 3–5, and ≥6 SCC in year prior, respectively
Lam 2021, USA	Number of transfusions	Post hoc of Niihara et al.	5–58, median 19 in treatment group, 17 in placebo	230	- 2.86 units transfused per year in treatment vs. 5.38 in placebo (*p* = 0.0253) - 1.702 RBC transfusion episodes per patient-year in the L-glutamine arm vs. 2.659 in placebo (*p* = 0.0783)
Wilson 2019, USA	Opioid use	Case series	Range 9–24	4	- All 4 patients had decreased opioid use, 21–100% decrease - No difference in average hemoglobin - 1 patient went from 3 ED visits + 2 hospitalizations to 0 in post-treatment period
*Side Effects and Prescribing Data*					
Gotesman 2020, USA	Prescribing data, compliance	Retrospective review	Mean 9.1	50	- 83% of patients 5y and older prescribed L-glutamine, 70% were dispensed - Some refused due to taste or abdominal pain - One-fourth of patients had <50% intake while on both HU and L-glutamine; overall compliance was ≥70% in 12 patients - Of 10 patients, mean number of crises decreased 0.9 to 0.2, *p* = 0.016
Ogu 2019, USA	Compliance, side effects	Retrospective review	Mean 36, range 21–70	111	- At end of 14-month period, 19% were actively taking L-glutamine, 42% discontinued, 35% never filled, 4% received but did not start - Mean fill rate: 1.79 fills, 2.51 for patients who filled more than once - 47% filled two times or fewer - Median day to stopping 47 days - Barriers to initiating: 38% prior authorization (PA) denied, 21% high deductible, 10% other insurance issues - Reasons to stopping: 35% poor adherence, 13% side effects, 4% pregnant, 4% no perceived benefit - GI side effects: nausea, abdominal pain, and constipation

**Table 2 pharmacy-10-00123-t002:** Summary of findings from included studies of crizanlizumab (Adakveo).

Source (Country)	Outcome Measured	Study Design	Age, Years	Group, *n*	Summary of Findings
*Clinical Efficacy*					
Ataga 2017, USA	Primary: rate of sickle-cell pain crises Secondary: hospitalizations, time to first and second hospitalizations, annual rates of uncomplicated crises, ACS, patient-reported outcomes	RCT (SUSTAIN trial)	Median 29, range 16–63	198	- Median rate of crises 1.63 with high-dose vs. 2.98 with placebo (45.3% lower, *p* = 0.01) - Median time to first crisis longer with high-dose crizanlizumab than with placebo (4.07 vs. 1.38 months, *p* = 0.001) - Median time to second crisis longer (10.32 vs. 5.09 months, *p* = 0.02) - Median rate of uncomplicated crises per year 1.08 with high-dose crizanlizumab, 2.91 with placebo (62.9% lower, *p* = 0.02) - Adverse events in 10% of either active treatment group and at least twice than in the placebo group: arthralgia, diarrhea, pruritus, vomiting, and chest pain
Ataga 2019, USA	Annual rate of hospitalization, time to first hospitalization	Post hoc of SUSTAIN data	Median 29, range 16–63	198	- 46% of patients in treatment group were not hospitalized vs. 35% in placebo arm - 54% in treatment group had ≥1 hospitalization vs. 65% in placebo arm - Median time to first hospitalization 6.3 in treatment vs. 3.2 months
Kutlar 2019, USA	Secondary endpoints of SUSTAIN	Post hoc of SUSTAIN data	Median 29, range 16–63	198	- % with no crises: 35.8% in 5/kg group crisis-free, 18.2% in 2.5 mg/kg, and 16.9% in placebo ○ 5–10 crises in last year: 28.0% vs. 4.2% ○ HbSS: 31.9% vs. 17.0% ○ Taking HU: 33.3% vs. 17.5% - In almost all subpopulations, crizanlizumab 5.0 mg/kg significantly (*p* < 0.05) increased time to first event by 2× or greater ○ 5–10 crises: 2.43 vs. 1.03 months (HR 0.47) ○ HbSS genotype: 3.7-fold increase (4.07 vs. 1.12 months; HR: 0.50) ○ HU use: 2.43 vs. 1.15 (HR 0.58)
Smith 2020, USA	Days of opioid use	Post hoc of SUSTAIN data	Median 29, range 16–63	198	- Absolute reduction of 4 days of opioid use in crizanlizumab vs. placebo (relative reduction: 57%, *p* = 0.162) - Reduction in IV use: absolute reduction 2.01 days (relative reduction: 50%, *p* = 0.047)
Shah 2019, USA	Pain crises events, treatment patterns, health care resources	Retrospective cohort review of SUSTAIN 1 year following study	≥18 years old, median 37	6	- Patients with placebo: 4 VOC events, w/2.5/kg: 5, with 5/kg: 0–2 - 4 patients with HU use during and after - All used opioids after - 2 patients had transfusions - 5 used healthcare resources, 1 did not and had 0 VOC events
*Side Effects and Prescribing Data*					
Kanter 2019, USA	Safety, side effects	Pooled Phase 2 trial data (SUSTAIN + SOLACE-adults) SUSTAIN: RCT SOLACE-adult: PK/PD	Median 29, range 16–65	111 (SUSTAIN *n* = 66, SOLACE *n* = 45)	- 84.7% had ≥1 adverse event - 21.6% had serious AEs, 5.4% thought to be due to crizanlizumab - Headache (19.8%), nausea (16.2%), back pain (15.3%) most common - 25.2% discontinued treatment prematurely - 45.9% had infection: 11.7% URI, 9.9% UTI - 1.8% infusion-related reaction - No clinically relevant laboratory (hematology, biochemistry, liver) or ECG abnormalities, or vital sign changes
Kanter 2021, USA	Insurance approval, adherence	Retrospective multi-center review	≥16	297 prescribed, 238 received infusion	- 12% denied by insurance - 32% discontinued therapy - 64% do not automatically use pre-medications - Reasons for discontinuing: lack of improvement or feeling pain was increased, transportation issues, infusion pain
Kanter 2021, USA	Infusion-related reaction	Retrospective review of safety database	Median 23 (range 16–38)	28	- From *Novartis* database, 28 patients had infusion-related reaction presenting as pain event, RR of 1.67 cases per 100 point-years - 86% experiences at 1st or 2nd infusion, majority recovered within 3 days - 6 (21%) had reactions on subsequent infusions - 71% were hospitalized for further treatment, 32% reported SCD complications after reaction (ACS, fat embolism, hemolytic crisis, pneumonia, multi-organ failure) - 82% had crizanlizumab discontinued after reaction occurrence
Li 2021, USA	Side effects	Case report	20 and 48	2	- 2 patients developed acute febrile painful episodes on 2nd infusion of crizanlizumab despite pre-medications

**Table 3 pharmacy-10-00123-t003:** Summary of findings from included studies of voxelotor (Oxbryta).

Source (Country)	Outcome Measured	Study Design	Age, Years	Group, *n*	Summary of Findings
*Clinical Efficacy*					
Vichinsky 2019, international	Primary: hemoglobin response	Phase 3 RCT (HOPE trial)	12–64, median 24	274	- 51% in 1500 mg had response vs. 7% in placebo (*p* < 0.001) - At week 24, significant reduction in baseline indirect bilirubin and reticulocyte count - Mean change in hemoglobin: 1.1 g/dL in 1500 vs. 0.6 in 900 vs. −0.1 in placebo (*p* < 0.001) - Markers of hemolysis: indirect bilirubin −29.1% vs. −3.2% (*p* < 0.001), reticulocytes −19.9% vs. +4.5% (*p* < 0.001) - Annual incidence of VOE: 2.77 vs. 2.76 vs. 3.19 - Similar adverse events across all groups: at least grade 3 in 26% in 1500, 23% in 900, 26% in placebo - Most common: headache (26%), diarrhea (20%), nausea (17%) in 1500 mg group - No substantial differences in the percentages of participants who had sickle-cell disease-related adverse events among the 3 groups
Howard 2019, international	Measurements of hemolysis	Post hoc of HOPE trial	12–64, median 24	274	- Patients with hemoglobin change >1 g/dL had greatest reduction in markers of hemolysis (reticulocyte count, indirect bilirubin, LDH) - Patients with hemoglobin change >1 g/dL had greater effect with 1500 vs. 900 mg
Howard 2021, international	Changes from baseline hemoglobin, safety	72-week follow-up of HOPE	12–64, median 24	274	- 89% receiving 1500 mg had hemoglobin increased of 1 g/dL or greater over 72 w vs. 25% of placebo (*p* < 0.001) - Mean change 1.0 g/dL vs. 0 (*p* < 0.001) - Improvements in hemolysis: indirect bilirubin, reticulocyte count, LDH - Lower number of VOEs but not statistically significant
Minniti 2021, international	Leg ulcers	Post hoc of HOPE	12–64, median 24	274	- Nearly all patients (>90%) receiving voxelotor (1500 and 900 mg) had their leg ulcers improve or resolve by week 72 - Resolution of leg ulcers was associated with increased Hb levels
Vichinsky 2020, international	Hemoglobin response and VOE incidence	72-week follow-up of HOPE	12–64, median 24	274	- 60 study participants achieved average Hb levels ≥10 g/dL and 10 achieved average Hb ≥12 g/dL - Incidence rate of VOCs was lowest in patients who achieved the highest Hb levels (≥12 g/dL) compared with those who achieved lower Hb levels and those receiving placebo
Achebe 2021, international	Hemoglobin response, markers of hemolysis	Long-term open-label extension of HOPE, 1500 mg	Median 25	178	- Patients who received placebo during HOPE: mean change in Hb 1.3 g/dL; if had already received voxelotor: 900 mg, 0.7 g/dL, if 1500 mg, 0.2 g/dL - Hemolysis: −39.5% indirect bilirubin in previous placebo recipients, −28.6% reticulocyte percentage; stable for those who received voxelotor during HOPE: −2.0% and 1.1% indirect bilirubin, −14.6% and −21.0% reticulocytes for voxelotor 900 mg and 1500 mg - 83.7% experiences non-SCD related AE, most grade 1 or 2 - 6.2% had AE leading to discontinuation
Estepp 2021, USA	Increase in hemoglobin, side effects	Open-label, multicenter, multiple-dose trial (HOPE KIDS 1)	4–11, median 7	45	- 84% on baseline HU - At week 24, mean change in hemoglobin was +1 g/dL - 47.1% achieved hemoglobin response - Decreases in markers of hemolysis: −28.6% indirect bilirubin, −2.6% LDH, −3.3% reticulocytes - 48.9% had side effects: diarrhea 11%, 11% vomiting, 11% rash - 9% discontinued due to side effects
Brown 2018, USA	Increase in hemoglobin	Phase 2a study of 900 mg/day in adolescents	12–17, median 14	25	- 92% on baseline HU - 42% had increase of hemoglobin >1 g/dL - Mean reduction 32% reticulocytes, 38% indirect bilirubin - No serious adverse events, no one terminated use - Most common side effects: nausea 12%, vomiting 8%, headache 8%, rash 8%
Brown 2019, USA	Increase in hemoglobin	Phase 2a study of 1500 mg/day in adolescents	12–17, median 14	15	- 100% on baseline HU - 55% had increase of hemoglobin >1 g/dL, median increase of 1.1 - Markers of hemolysis: −5.8% reticulocytes, −36.9% indirect bilirubin, −23.1% LDH - Most side effects were grade 1 or 2, one grade 3 (rash), no one discontinued
Muschick 2021, USA	Increase in hemoglobin	Single-center retrospective review	12–21	17	- 100% on baseline HU - Hemoglobin improved mean 1.49 g/dL - Reticulocyte percentage decreased 4.14%, total bilirubin by 1.71 mg/dL - All reported subjective clinical improvement (increased energy, decreased pain) - Side effects: mild diarrhea, nausea, necessitating supportive care
Phan 2021, USA	Exercise capacity	Pilot study	12–20	9	- Mean rise in hemoglobin +1.3 g/dL, decrease reticulocyte count −2.4%, bilirubin −0.4 mg/dL - Changes in peak VO2 ranged from −10% to 10%, not statistically significant - 7 of 9 with subjective positive change on lifestyle questionnaire
Zaidi 2020, USA	Effect on anemia	Retrospective review in Symphony Health system	Mean 35.7	1275	- *n* = 52 had hemoglobin measurements following initiation of voxelotor - Mean increase in Hb from baseline was 1.1–1.3 g/dL - 55% achieved increase of >1 g/dL of hemoglobin 1 year - Decrease in transfusion rates (*p* < 0.05) - Decrease in VOC rates (*p* = 0.258)
*Side Effects and Prescribing Data*					
Ware 2020, international	Concomitant use of HU	Post hoc of HOPE	12–64, median 24	274	- 65% of patients receiving HU at study enrollment - Similar percentages of participants reported treatment-emergent adverse events in those receiving HU and those not - Voxelotor led to increase in hemoglobin levels but not MCV, %HbF, and ANC, consistent with stable HU exposure throughout the study
Betancourt 2020, USA	Real-world experience, barriers to use, health outcomes	Single-center retrospective review	21–70, mean 41	54	- 52% also taking HU at start of voxelotor - After 2–4 weeks, 63% had increase of hemoglobin of >1 g/dL - 65% able to access drug, mean time to obtain drug 45 days - Barriers: prior authorization or appeal 9%, incomplete follow-up 7%, missing documentation 6%, and high patient co-pay 6% - Side effects: GI 23%, dermatologic 17%, increase in pain 11% - 29% required dose modification from side effect, 23% terminated medication
Shah 2020, USA	Prescribing, side effects, clinical efficacy	Multicenter retrospective review	Mean 33	60	- 80% on HU, 20% on chronic transfusions, 10% on Epo-stimulating agents, average hemoglobin 7.4 g/dL - 52% were prescribed but did not start taking or did not follow-up for labs - 15% required dose adjustment to 1000 mg due to side effects - Average hemoglobin at 1 month 8.6 g/dL, at 3 months 8 g/dL - 52% increased by 1 g/dL at 1 month and 44% increased by 1 g/dL at 3 months - Increased report of more energy, quality of life - 5% discontinued due to pregnancy, transaminitis, GI symptoms
Andemariam 2021, USA	Prescribing, side effects, clinical efficacy	Multicenter retrospective post-marketing review	Mean 34.3	300, 49 at time of data analysis	- Rationale for prescription: reduction of anemia (73.5%), reduction in frequency of VOEs (46.9%), reduction in pain (69.4%), reduction in the need for blood transfusion (16.3%) - Average 1.6 g/dL increase of hemoglobin from baseline - 50% had >1 g/dL improvement - Improvement in hemolytic markers: −4.9% reticulocyte percentage, −1.9 mg/dL indirect bilirubin - 38.8% reported at least one adverse event, most common diarrhea, headache, rash, most mild

## Data Availability

No new data were created or analyzed in this study. The data presented in this study are available on request from the corresponding author.

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
