# Peer review of "Recent Advances in Sickle-Cell Disease Therapies: A Review of Voxelotor, Crizanlizumab, and L-glutamine"

_pharmacy, 2022, doi:10.3390/pharmacy10050123_

Round 1

Reviewer 1 Report

11.      In abstract, propose to include below quoted concluding sentence at the end of abstract to clearly present aim of this review, meaning- author's point of view regarding purpose for this review to readers.

“Aim of the article to evaluate post-approval studies of crizanlizumab, voxelotor, and L-glutamine in Sickle Cell Disease (SCD), with a focus on real-world efficacy, side effects, and prescribing data.”

22.      Discrepancy in the number articles included for voxelotor in the abstract and in section 2.1/Figure 1.  Abstract mentioned 15 articles and section 2.1/Figure 1 it was stated 16 articles for voxelotor. Please check.

Author Response

Reviewer 1

Comment: In abstract, propose to include below quoted concluding sentence at the end of abstract to clearly present aim of this review, meaning- author's point of view regarding purpose for this review to readers. “Aim of the article to evaluate post-approval studies of crizanlizumab, voxelotor, and L-glutamine in Sickle Cell Disease (SCD), with a focus on real-world efficacy, side effects, and prescribing data.”

Response: We appreciate the reviewer’s comment and agree with this suggestion. We added the suggested statement from the reviewer to the abstract.

Comment: Discrepancy in the number articles included for voxelotor in the abstract and in section 2.1/Figure 1.  Abstract mentioned 15 articles and section 2.1/Figure 1 it was stated 16 articles for voxelotor. Please check.

Response: We appreciate the reviewer’s comment, and we apologize for this oversight. The number of voxelotor studies in the abstract has been corrected from 15 to 16.

Reviewer 2 Report

In this paper from Migotsky et al, the aim was to evaluate post-approval studies focusing on efficacy, side effects and prescription data. Main contribution is a concise and complete review of post-approval data, including real world data to the literature. Manuscript is clearly written and relevant to the field of sickle cell disease. Comments below:

1. Line 70: adjust grammar. Omit "was performed".

2. Line 100: Table 1 is referenced in the text but no table provided.

3. Line 134: Typo. Should read "real world".

Author Response

In this paper from Migotsky et al, the aim was to evaluate post-approval studies focusing on efficacy, side effects and prescription data. Main contribution is a concise and complete review of post-approval data, including real world data to the literature. Manuscript is clearly written and relevant to the field of sickle cell disease. Comments below:

Comment: Line 70: adjust grammar. Omit "was performed".

Response: We appreciate the reviewer’s comment, and we apologize for this oversight. We removed “was performed” from the text.

Comment: Line 100: Table 1 is referenced in the text but no table provided.

Response: We appreciate the reviewer’s comment. Table 1 included as a separate file.

Comment: Line 134: Typo. Should read "real world".

Response: We appreciate the reviewer’s comment, and we apologize for this oversight. We corrected “real-work” to “real-world”.